# LEARNING SPATIO-TEMPORAL REPRESENTATION FOR MULTIVARIATE TIME SERIES

## ABSTRACT

Label sparsity renders the use of the label information of multivariate time series (MTS) challenging in practice. Thus, unsupervised representation learning methods have been studied to learn effective representations of MTS without label information. Recently, many studies have employed contrastive learning to generate robust representations by capturing underlying information about MTS. However, they have some limitations, such as the insufficient consideration of relationships between the variables of MTS and high sensitivity to positive pairs. We proposed a novel spatio-temporal contrastive representation learning method (STCR) for generating effective MTS representations suitable for classification and forecasting tasks. STCR learns representations by encouraging spatio-temporal consistency, which comprehensively reflects the spatial information and temporal dependency of MTS and simultaneously mitigates sensitivity to constructing positive pairs for contrastive learning. The results of extensive experiments on MTS classification and forecasting tasks demonstrate the efficacy of STCR in generating high-quality representations and state-of-the-art performance on both tasks.

## 1 INTRODUCTION

Multivariate time series (MTS), which consists of synchronous variables related to one another over time, is a crucial data type that is used in various fields, such as engineering, finance, and medicine (Ding et al., 2022; Lee et al., 2023). However, using MTS label information in practice is challenging because MTS is sometimes sparsely labeled (Ching et al., 2018; Wickstrøm et al., 2022). Therefore, learning universal representation, which is suitable for various tasks involving MTS, without label information has attracted considerable attention (Jing & Tian, 2020; Cheng et al., 2021).

Contrastive learning has achieved superior performance in generating representations without label information in a self-supervised manner (Zhang & Ma, 2022; Xu et al., 2022). This approach encourages the representations of context views, which are the variants of the original data with transformations, especially data augmentations, to be similar; thereby, effective representations can be learned, achieving promising performance in various downstream tasks. However, the unique characteristics of time series, such as temporal dependency and irregular patterns, have hindered the application of conventional augmentation-based contrastive learning approaches (Wickstrøm et al., 2022). For example, applying rotation, one of the augmentation methods typically used in the computer vision domain, to time series may corrupt their trends or patterns because of the change in the distribution (Yue et al., 2022). Thus, contrastive learning methods specialized in time series have emerged by introducing augmentation methods suitable for time series or effectively reflecting temporal structural information (Yang et al., 2022; Yue et al., 2022). Although these methods can generate useful representations for various tasks of time series, two notable limitations exist.

First, in most of these methods, spatial information in MTS is not considered. Spatial information, which indicates structural relations between MTS variables, is crucial in various tasks because it enables the representations to reflect the information of associated variables (Yu et al., 2017; Guo et al., 2021), that is, spatial information can improve classification performance by enabling representations with similar structures to be close (Kadous & Sammut, 2005; Cini et al., 2021). Therefore, the existing methods that do not consider spatial information cannot achieve high performance in tasks for MTS compared with those for univariate time series (UTS).

Second, the representations obtained by time-series contrastive learning are sensitive to configured positive pairs (Yue et al., 2022). Three typical selection strategies exist to construct positive pairs:

1. *Subseries consistency* (Franceschi et al., 2019) constructs a time-series instance and its sampled subseries as a positive pair.

2. *Local consistency* (Tonekaboni et al., 2021) enforces the local smoothness of representations by selecting neighboring segments as positive samples.

3. *Transformation consistency* (Eldele et al., 2021) regards the augmented time series with specific transformation as a positive sample.

However, the representations obtained from these selection strategies are vulnerable to distribution changes in time series (Yue et al., 2022). For example, the subseries and local consistencies are vulnerable to level shifts and anomalies in time series, respectively. Thus, the existing methods may fail to configure proper positive and negative pairs, which causes performance degradation.

To address these problems, we proposed a **S**patio-**T**emporal **C**ontrastive **R**epresentation learning (STCR) that generates effective representation by capturing the inherent structure of MTS with spatio-temporal information. Particularly, we introduced spatio-temporal consistency that comprehensively considers the spatio-temporal relations in MTS and its temporal dependency while mitigating the drawbacks of the conventional selection strategies. Then, we learned effective representations of MTS by encouraging spatio-temporal consistency.

To demonstrate the effectiveness of STCR, we conducted extensive comparative experiments on classification and forecasting, which are major tasks in MTS, with state-of-the-art methods (SOTAs). Consequently, STCR generates remarkably informative representations that can achieve superior performances on both classification and forecasting tasks compared to those of SOTAs.

This study has the following contributions:

- We proposed a novel contrastive representation learning method that encourages *spatio-temporal consistency* to reflect spatial structures of MTS and its temporal dependency.

- To capture the inherent spatial information of MTS, STCR converts an MTS into graphs with diverse edge structures and enforces the representations from them to be consistent.

- Our method alleviates the drawbacks of the selection strategies for configuring positive pairs in time series by obtaining robust representations with spatio-temporal consistency.

- The proposed method generates universal representations that achieve superior performance for both classification and forecasting tasks compared with that of SOTAs.

## 2 RELATED WORKS

Because MTS datasets often have insufficient label information, unsupervised representation learning for MTS has been studied extensively (Wickstrøm et al., 2022). Generally, in existing methods, temporal dependency is considered for generating MTS representations. For example, T-Loss (Franceschi et al., 2019) explicitly considered the temporal structure of MTS using a triplet loss with time-based negative sampling to handle subseries consistency. TNC (Tonekaboni et al., 2021) captured local temporal relationships between timestamps by constructing a graph, in which each timestamp is considered to be a node, and learning a representation of the graph. Moreover, TST (Zerveas et al., 2021), a transformer-based representation learning method for MTS, was introduced.

Recent studies on MTS representation learning have used contrastive learning that effectively captures underlying information in time series. In TS-TCC (Eldele et al., 2021), several data augmentations, such as jittering, scaling, and permutation, were exploited to learn transformation-invariant representations of UTS and MTS. TS2Vec (Yue et al., 2022) introduced a hierarchical contrasting method to learn contextual representations for arbitrary subseries at various semantic levels.

However, the existing methods have some limitations. First, although spatial information is crucial to analyze MTS because its variables affect each other (Guo et al., 2021), most studies have focused on reflecting temporal information, not considering spatial information. Thus, their performance on MTS is low compared with that on UTS. Second, they used only one or two selection strategies,

vulnerable to certain distribution changes of MTS, to construct positive and negative pairs; thus, each method may inherit the drawback from the corresponding selection strategy (Yue et al., 2022).

By contrast, the proposed method effectively handles spatial information capturing relationships between variables as well as the temporal dependency of MTS, even alleviating the drawbacks of selection strategies by encouraging spatio-temporal consistency. Using STCR, we generated a universal representation that improves performance for both MTS classification and forecasting tasks.

# 3 PROPOSED METHOD

## 3.1 PROBLEM STATEMENT

Let $\mathcal{X} = \{x_i \in \mathbb{R}^{L_x \times V}\}_{i=1}^N$ be a set of MTS, where $L_x$ and $V$ are the sequence length and the number of variables, respectively. We define nonlinear mapping functions $f : x_i \rightarrow z_i^{\mathcal{T}}$, $g : x_i \rightarrow \mathcal{S}_i$, and $h : [z_i^{\mathcal{T}}; z_i^{\mathcal{S}}] \rightarrow z_i$, where $z_i^{\mathcal{T}}, \mathcal{S}_i$, and $z_i$ have dimensions $L_x \times d_{z^{\mathcal{T}}}, V \times d_{z^{\mathcal{S}}}$, and $L_x \times d_z$, respectively, and $z_i^{\mathcal{S}} \in \mathbb{R}^{L_x \times d_{z^{\mathcal{S}}}}$ is obtained by multiplying $x_i$ with $\mathcal{S}_i$. The objective is simultaneously learning $f$, $g$, and $h$ to map $x_i$ to its spatio-temporal representation $z_i$.

## 3.2 SPATIO-TEMPORAL CONTRASTIVE REPRESENTATION LEARNING

To learn an effective representation $z_i$ of an MTS instance $x_i$, we proposed *spatio-temporal consistency* that can simultaneously consider the spatial relations of MTS along with its temporal structure while alleviating the drawbacks of three typical selection strategies for constructing positive pairs. STCR simultaneously trains $f$, $g$, and $h$ by encouraging this consistency between two context views of $x_i$ obtained from four modules: *random cropping*, *temporal embedding*, *spatial embedding*, and *projection*. Figure 1 displays an overview of the proposed method for generating the representation of an MTS instance.

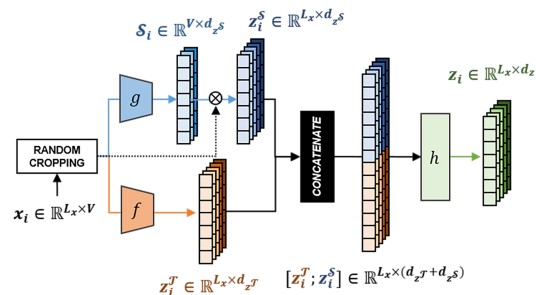

Figure 1: Overview of STCR

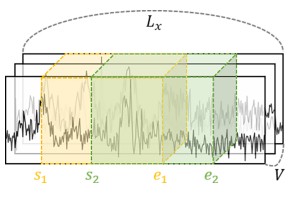

Figure 2: Random cropping

**Random cropping.** We randomly cropped an MTS instance to create two subseries, where each subseries is used for obtaining a context view. As shown in Figure 2, given an MTS instance $x_i \in \mathbb{R}^{L_x \times V}$, we randomly extracted two subseries $\tilde{x}_i$ and $\tilde{x}_i'$, which have overlapping time segments $[s_1, e_1]$ and $[s_2, e_2]$ such that $0 < s_1 \leq s_2 \leq e_1 \leq e_2 \leq L_x$. Following Franceschi et al. (2019), the learned representations on the overlapped segment $[s_2, e_1]$ should be consistent for two context views. This approach enables us to learn position-agnostic representations while avoiding dimension collapse (Yue et al., 2022). Note that random cropping is only used in the training phase.

**Temporal embedding.** As in Figure 3(a), to capture temporal structures of MTS, we first applied random masking to the subseries and subsequently obtained temporal features by passing the masked subseries through encoder $f$. Random masking helps to generate a transformation-invariant representation capturing the underlying temporal structure for MTS without a strong inductive bias (Yue et al., 2022). Let $\tilde{x}_i \in \mathbb{R}^{L_{\tilde{x}_i} \times V}$ be a subseries derived by random cropping for an MTS instance $x_i$. Next, $\tilde{x}_i$ is masked along the time axis with a binary mask $m_i \in \{0, 1\}^{L_{\tilde{x}_i}}$ that is independently sampled from a Bernoulli distribution with $p$ in every forward pass of a learning process. Note that the same mask vector is applied to every variable of MTS to focus on temporal consistency. Subsequently, the masked subseries is passed to encoder $f$ to derive temporal features $\tilde{z}_i^{\mathcal{T}} \in \mathbb{R}^{L_{\tilde{x}_i} \times d_{z^{\mathcal{T}}}}$ by

$$\tilde{z}_i^{\mathcal{T}} = f(m_i \times \tilde{x}_i). \tag{1}$$

The encoder $f$ has several temporal blocks that consist of one-dimensional dilated convolutional layers (*DilatedConv*) and *GeLU* activation functions for capturing the long-term dependency of MTS as a large receptive field (Bai et al., 2018).

**Spatial embedding.** In this module illustrated in Figure 3(b), we used a $K$-nearest neighbor ($K$-NN) graph, which effectively identifies the relations between the variables of MTS (Ferreira & Zhao, 2016). First, we calculated similarities between variables in $\tilde{x}_i$ using the heat kernel, a popular method used to construct the edges of $K$-NN graph (Bo et al., 2020) as follows:

$$\zeta_i^{(v,u)} = e^{-\|\tilde{x}_{i,v} - \tilde{x}_{i,u}\|^2/2}, \tag{2}$$

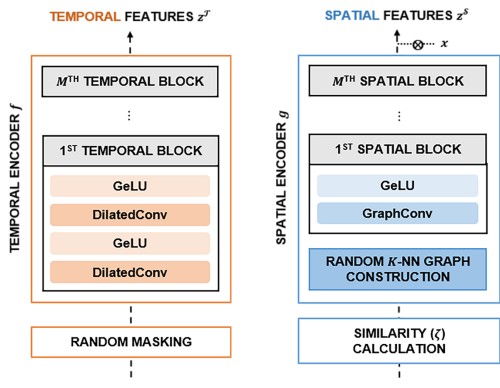

(a) Temporal Embedding     (b) Spatial Embedding

Figure 3: Two embedding modules

where $v$ and $u$ denote variables of $\tilde{x}_i$. Then, we converted $\tilde{x}_i$ into a $K$-NN graph, $\mathcal{G}_i^K$, where each node corresponds to a variable and the $K$ largest similarities for each variable form the connected edges. Next, we constructed an encoder $g$ consisting of spatial blocks with graph convolutional layers (*GraphConv*) followed by *GeLU* to handle the graph $\mathcal{G}_i^K$. However, because various graphs can be constructed depending on $K$, the graphs with different $K$s may contain different spatial information (Deng & Hooi, 2021). Therefore, when we convert MTS into $K$-NN graphs, setting the appropriate $K$ is a challenging problem. To recognize the underlying spatial structure of MTS by addressing this problem, we encouraged the representations of $K$-NN graphs to be consistent across various $K$. In particular, we randomly selected $K$ from $[\lceil \rho \times V \rceil, V]$ to construct $\mathcal{G}_i^K$, where $\rho$ is a *connection parameter*. The number of connections $K$ is independently sampled in every forward pass of a learning process. Subsequently, we used an adjacency matrix $A_i^K$ of $\mathcal{G}_i^K$ to capture spatial information $\mathcal{S}_i \in \mathbb{R}^{V \times d_z S}$ using the following expression:

$$\mathcal{S}_i = g(\tilde{x}_i, A_i^K). \tag{3}$$

To reflect the information of the node itself and stable learning of *GraphConv*, we used self-loop and feature normalization techniques to $A_i^K$ (Pham & Yang, 2010). Then, equation 3 is formulated as

$$\mathcal{S}_i = g(\tilde{x}_i, \bar{D}^{-\frac{1}{2}} \bar{A}_i^K \bar{D}^{-\frac{1}{2}}), \tag{4}$$

where $\bar{A}_i^K$ is an adjacency matrix with self-loop ($\bar{A}_i^K = A_i^K + I$), $I$ is an identity diagonal matrix of $A_i^K$, and $\bar{D}$ is a degree matrix of $\bar{A}_i^K$ ($\bar{D}_{ii} = \sum_j \bar{A}_{ij}^K$). However, $\mathcal{S}_i$ has no information about temporal structure; hence, we used $\mathcal{S}_i$ as a weight to allow $\tilde{x}_i$ to reflect spatial relations between variables over time. We multiplied $\tilde{x}_i$ with $\mathcal{S}_i$ to generate a feature vector $\tilde{z}_i^{\mathcal{S}} \in \mathbb{R}^{L_{\tilde{x}_i} \times d_z S}$ as follows:

$$\tilde{z}_i^{\mathcal{S}} = \tilde{x}_i \times \mathcal{S}_i = \tilde{x}_i \times g(\tilde{x}_i, \bar{D}^{-\frac{1}{2}} \bar{A}_i^K \bar{D}^{-\frac{1}{2}}). \tag{5}$$

**Projection.** To obtain a spatio-temporal feature vector $\tilde{z}_i$, one of the context views of $x_i$, we concatenated the temporal and spatial feature vectors and passed them through a projection head $h$:

$$\tilde{z}_i = h\left([\tilde{z}_i^{\mathcal{T}}; \tilde{z}_i^{\mathcal{S}}]\right). \tag{6}$$

**Loss function.** To simultaneously train $f$, $g$, and $h$, we encouraged the *spatio-temporal consistency* of two context views obtained from the same MTS instance using two contrastive loss functions: *instance-wise* and *timestamp-wise*. Given two subseries, $\tilde{x}_i$ and $\tilde{x}_i'$, randomly cropped from $x_i$, we obtained $\tilde{z}_i$ and $\tilde{z}_i'$ by passing each $\tilde{x}_i$ and $\tilde{x}_i'$ through the temporal embedding, spatial embedding, and projection modules. Next, we used representations from other MTS instances at timestamp $t$ in the same batch as negatives to calculate instance-wise loss $\mathcal{L}_i^{IW}$ using the following equation:

$$\mathcal{L}_i^{IW} = -\frac{1}{L_x} \sum_{t=1}^{L_x} log \frac{e^{\tilde{z}_{i,t} \cdot \tilde{z}_{i,t}'}}{\sum_{j=1}^{B} \left(e^{\tilde{z}_{i,t} \cdot \tilde{z}_{j,t}'} + \mathbb{1}_{[i \neq j]} e^{\tilde{z}_{i,t} \cdot \tilde{z}_{j,t}}\right)}, \tag{7}$$

where $B$ is the batch size, and $\mathbb{1}$ is an indicator function. However, although the instance-wise contrastive loss is effective for the classification task, it is insufficient for the forecasting task, which requires fine-grained representations for every timestamp (Yue et al., 2022). Thus, we used a timestamp-wise contrastive loss function to obtain a discriminate representation over time for achieving a decent performance in forecasting as well as classification. For the timestamp-wise contrastive loss function, STCR defines the representations at the same timestamp from two context views of $x_i$ as positive pairs, whereas those at different timestamps from $x_i$ are defined as negative pairs; thereby, the timestamp-wise contrastive loss $\mathcal{L}_i^{TW}$ is formulated as follows:

$$\mathcal{L}_i^{TW} = -\frac{1}{L_x} \sum_{t=1}^{L_x} log \frac{e^{\tilde{z}_{i,t} \cdot \tilde{z}'_{i,t}}}{\sum_{t' \in \Phi} \left( e^{\tilde{z}_{i,t} \cdot \tilde{z}'_{i,t'}} + \mathbb{1}_{[t \neq t']} e^{\tilde{z}_{i,t} \cdot \tilde{z}_{i,t'}} \right)}, \tag{8}$$

where $\Phi$ is the set of the overlapping timestamps of two subseries. Finally, the overall loss $\mathcal{L}$ of STCR is calculated as follows:

$$\mathcal{L} = \frac{1}{N} \sum_{i=1}^{N} \left( \mathcal{L}_i^{IW} + \mathcal{L}_i^{TW} \right), \tag{9}$$

where $N$ is the number of MTS instances in the training dataset. Through this learning process, we can obtain $f$, $g$, and $h$ for generating universal representations suitable for various tasks of MTS. The proposed method is summarized in Algorithm 1.

---

**Algorithm 1** Learning procedure of STCR

---

**Input:** MTS dataset $\mathcal{X} = \{x_1, x_2, \cdots, x_N\}$, temporal encoder $f$, spatial encoder $g$, projection head $h$, and number of optimization iterations $Q$
**Output:** Trained $f$, $g$, and $h$
    Initialize $f$, $g$, and $h$.
    **while** $q \leq Q$ **do**
        **for** $x_i \in \mathcal{X}$ **do**
            Randomly crop $x_i$ to overlapped subseries $\tilde{x}_i$ and $\tilde{x}'_i$.
            Create two context views $\tilde{z}_i$ and $\tilde{z}'_i$ by equations 1-6.
            Calculate $\mathcal{L}_i^{IW}$ by equation 7.
            Calculate $\mathcal{L}_i^{TW}$ by equation 8.
        **end for**
        Update $f$, $g$, and $h$ by equation 9.
    **end while**

---

Furthermore, we analyzed the complexity of the proposed method from two perspectives of the number of learnable parameters and computational time in Section C.1.

## 4 EXPERIMENTAL RESULTS

We evaluated the representations generated by STCR on classification and forecasting tasks of MTS. Here, the experimental results for each task are described in detail. The implementation detail of STCR is provided in Section A. Also, the experimental settings for each task are given in Section B.

### 4.1 MULTIVARIATE TIME-SERIES CLASSIFICATION

Table 1 depicts the classification performance of the proposed method compared with the baseline methods for 11 MTS datasets. Moreover, we performed statistical tests on the classification performance to ensure the significance of performance improvement by STCR. In particular, we used a two-sample Wilcoxon signed rank test (Conover, 1999) between STCR and each baseline method. The superscripts * and ** imply the rank test's p-value was smaller than 0.1 and 0.05, respectively.

STCR achieved the best performance in six of 11 datasets, and the average rank was 2.091, outperforming the baselines. Moreover, the statistical tests showed that STCR performed significantly better than SOTAs. Notably, STCR achieved overwhelming performance on MTS datasets regardless of the number of variables $V$ and sequence length $L_x$ by reflecting both temporal and spatial

Table 1: Classification performance for STCR compared with baselines. For each dataset, the best accuracy score is highlighted in boldface.

| Dataset | $V$ | $L_x$ | Method | | | | | | |
|---|---|---|---|---|---|---|---|---|---|
| | | | DTW | T-Loss | TNC | TS-TCC | TST | TS2Vec | **STCR** |
| HandMovementDirection | 10 | 400 | 0.231 | **0.351** | 0.324 | 0.243 | 0.243 | 0.338 | **0.351** |
| PhonemeSpectra | 11 | 217 | 0.151 | 0.222 | 0.207 | **0.252** | 0.085 | 0.233 | 0.186 |
| JapaneseVowels | 12 | 29 | 0.949 | **0.989** | 0.978 | 0.930 | 0.978 | 0.984 | **0.989** |
| SpokenArabicDigits | 13 | 93 | 0.963 | 0.905 | 0.934 | 0.970 | 0.923 | **0.988** | 0.955 |
| NATOPS | 24 | 51 | 0.883 | 0.917 | 0.911 | 0.822 | 0.850 | 0.928 | **0.944** |
| FingerMovements | 28 | 50 | 0.530 | 0.580 | 0.470 | 0.460 | 0.560 | 0.480 | **0.590** |
| Heartbeat | 61 | 405 | 0.717 | 0.741 | 0.746 | **0.751** | 0.746 | 0.683 | 0.746 |
| MotorImagery | 64 | 3000 | 0.500 | 0.580 | 0.500 | **0.610** | 0.500 | 0.510 | 0.550 |
| FaceDetection | 144 | 62 | 0.529 | 0.513 | 0.536 | 0.544 | 0.534 | 0.501 | **0.557** |
| InsectWingbeat | 200 | 30 | - | 0.156 | **0.469** | 0.264 | 0.105 | 0.466 | 0.453 |
| PEMS-SF | 963 | 144 | 0.711 | 0.676 | 0.699 | 0.734 | 0.740 | 0.682 | **0.931** |
| Average Rank | | | 5.273** | 3.818** | 3.909** | 3.636* | 4.545** | 3.818* | **2.091** |

information of MTS. For example, for the *PEMS-SF* dataset, which $V = 963$ and $L_x = 144$, STCR improved classification performance by approximately 19% than the second-best accuracy score.

By contrast, the baselines exhibited performance differences according to the number of variables or the sequence length. The second-best method in average rank, TS-TCC, performed poorly in the datasets with short sequences, such as *JapanesVowels* and *InsectWingbeat*. In addition, T-Loss and TS2Vec showed the worst performance in the datasets with many variables, including *PEMS-SF* and *FaceDetection*. These results confirmed that STCR is effective for MTS classification tasks.

## 4.2 MULTIVARIATE TIME-SERIES FORECASTING

Table 2 shows the forecasting results of STCR compared with the baselines of forecasting tasks.

Table 2: Forecasting results on MSE and MAE for STCR compared to the baselines. Here, $H$ denotes the prediction length. For each dataset, the lowest MSE and MAE are highlighted in boldface.

| Dataset | $H$ | Informer | | StemGNN | | TCN | | LogTrans | | LSTnet | | TS2Vec | | **STCR** | |
|---|---|---|---|---|---|---|---|---|---|---|---|---|---|---|---|
| | | MSE | MAE | MSE | MAE | MSE | MAE | MSE | MAE | MSE | MAE | MSE | MAE | MSE | MAE |
| ETTh1 | 24 | **0.577** | 0.549 | 0.614 | 0.571 | 0.767 | 0.612 | 0.686 | 0.604 | 1.293 | 0.901 | 0.599 | **0.534** | 0.591 | 0.542 |
| | 48 | 0.685 | 0.625 | 0.748 | 0.618 | 0.713 | 0.617 | 0.766 | 0.757 | 1.456 | 0.960 | 0.629 | **0.555** | **0.626** | 0.565 |
| | 168 | 0.931 | 0.752 | **0.663** | **0.608** | 0.995 | 0.738 | 1.002 | 0.846 | 1.997 | 1.214 | 0.755 | 0.636 | 0.762 | 0.644 |
| | 336 | 1.128 | 0.873 | 0.927 | 0.730 | 1.175 | 0.800 | 1.362 | 0.952 | 2.655 | 1.369 | **0.907** | **0.717** | 0.910 | 0.721 |
| | 720 | 1.251 | 0.896 | - | - | 1.453 | 1.311 | 1.397 | 1.291 | 2.143 | 1.380 | 1.048 | 0.790 | **0.997** | **0.770** |
| ETTh2 | 24 | 0.720 | 0.665 | 1.292 | 0.883 | 1.365 | 0.888 | 0.828 | 0.750 | 2.742 | 1.457 | 0.398 | **0.461** | 0.375 | 0.467 |
| | 48 | 1.457 | 1.001 | 1.099 | 0.847 | 1.395 | 0.960 | 1.806 | 1.034 | 3.567 | 1.687 | **0.580** | **0.573** | 0.605 | 0.597 |
| | 168 | 3.489 | 1.515 | 2.282 | 1.228 | 3.166 | 1.407 | 4.070 | 1.681 | 3.242 | 2.513 | 1.901 | 1.065 | **1.587** | **1.002** |
| | 336 | 2.723 | 1.340 | 3.086 | 1.351 | 3.256 | 1.481 | 3.875 | 1.763 | 2.544 | 2.591 | 2.304 | 1.215 | **1.956** | **1.130** |
| | 720 | 3.467 | 1.473 | - | - | 3.690 | 1.588 | 3.913 | 1.552 | 4.625 | 3.709 | 2.650 | 1.373 | **2.092** | **1.159** |
| ETTm1 | 24 | **0.323** | **0.369** | 0.620 | 0.570 | 0.324 | 0.374 | 0.419 | 0.412 | 1.968 | 1.170 | 0.443 | 0.436 | 0.430 | 0.432 |
| | 48 | 0.494 | 0.503 | 0.744 | 0.628 | **0.477** | **0.450** | 0.507 | 0.583 | 1.999 | 1.215 | 0.582 | 0.515 | 0.560 | 0.514 |
| | 96 | 0.678 | 0.614 | 0.709 | 0.624 | 0.636 | 0.602 | 0.768 | 0.792 | 2.762 | 1.542 | 0.622 | 0.549 | **0.601** | **0.545** |
| | 288 | 1.056 | 0.786 | 0.843 | 0.683 | 1.270 | 1.351 | 1.462 | 1.320 | 1.257 | 2.076 | 0.709 | 0.609 | **0.665** | **0.589** |
| | 672 | 1.192 | 0.926 | - | - | 1.381 | 1.467 | 1.669 | 1.461 | 1.917 | 2.941 | 0.786 | 0.655 | **0.779** | **0.654** |
| Electricity | 24 | 0.312 | 0.387 | 0.439 | 0.388 | 0.305 | 0.384 | 0.297 | 0.374 | 0.356 | 0.419 | **0.287** | **0.374** | 0.331 | 0.406 |
| | 48 | 0.392 | 0.431 | 0.413 | 0.455 | 0.317 | 0.392 | 0.316 | 0.389 | 0.429 | 0.456 | **0.307** | **0.388** | 0.352 | 0.421 |
| | 168 | 0.515 | 0.509 | 0.506 | 0.518 | 0.358 | 0.423 | 0.426 | 0.466 | 0.372 | 0.425 | **0.332** | **0.407** | 0.375 | 0.438 |
| | 336 | 0.759 | 0.625 | 0.647 | 0.596 | **0.349** | 0.416 | 0.365 | 0.417 | 0.352 | **0.409** | 0.349 | 0.420 | 0.391 | 0.449 |
| | 720 | 0.969 | 0.788 | - | - | 0.447 | 0.486 | **0.344** | **0.403** | 0.380 | 0.443 | 0.375 | 0.438 | 0.414 | 0.467 |
| Average | | 1.156 | 0.781 | 0.977 | 0.706 | 1.192 | 0.837 | 1.314 | 0.892 | 1.903 | 1.444 | 0.828 | 0.636 | **0.770** | **0.626** |

* All $H \geq 672$ cases of StemGNN fail for the out-of-memory even when $B = 1$.

STCR achieved the lowest average MSE and MAE of 0.770 and 0.626, respectively. STCR outperformed the baselines on long-term forecasting with long prediction length $H$ because spatial information between variables of MTS enhances the ability to recognize long-term patterns (Guo et al., 2021), and *DilatedConv* can capture long-term dependency (Li & Zhu, 2021). However, the forecasting performance of STCR for *Electricity* was slightly low. This dataset has only two variables,

and even one of the two variables is timestamp information. Although we used additional timestamp features through preprocessing, these variables may not have sufficient structural relations among them. Thus, capturing the inherent relationship between variables may not be necessary. However, STCR showed performance comparable to several baseline methods, even for the *Electricity* dataset.

## 4.3 ABLATION STUDIES

We performed extensive ablation studies to demonstrate the effectiveness of each component of STCR. STCR creates two context views using three *context components*: 1) random cropping, 2) random masking in the temporal embedding module, and 3) randomness on $K$ in the spatial embedding module. The representation is obtained by encouraging spatio-temporal consistency with two *contrastive loss functions*: 1) instance-wise and 2) timestamp-wise. We compared our approach to three ablation models for context components: STCR without random cropping (*STCR w/o C*), STCR without random masking (*STCR w/o M*), and STCR without randomness

Table 3: Accuracy scores of ablation models and STCR. For each dataset, the best score is highlighted in boldface. (APDR: Average Performance Drop Rate)

| Dataset | Context Component | | | Loss Function | | STCR |
|---|---|---|---|---|---|---|
| | *w/o C* | *w/o M* | *w/o K* | *w/o $\mathcal{L}^{IW}$* | *w/o $\mathcal{L}^{TW}$* | |
| Hand. | **0.392** | 0.311 | 0.284 | 0.270 | 0.351 | 0.351 |
| Phoneme. | 0.189 | **0.209** | 0.182 | 0.163 | 0.153 | 0.186 |
| Japaness. | 0.976 | 0.984 | 0.984 | 0.965 | 0.981 | **0.989** |
| Spoken. | **0.960** | 0.930 | 0.952 | 0.952 | 0.935 | 0.955 |
| NATOPS | 0.917 | 0.922 | 0.911 | 0.906 | 0.900 | **0.944** |
| Finger. | 0.520 | 0.480 | 0.470 | 0.480 | 0.520 | **0.590** |
| Heart. | 0.717 | 0.722 | 0.722 | 0.737 | 0.727 | **0.746** |
| Motor. | 0.500 | 0.500 | 0.490 | 0.500 | 0.520 | **0.550** |
| Face. | 0.551 | 0.526 | 0.528 | 0.549 | 0.522 | **0.557** |
| Insect. | 0.449 | 0.449 | 0.441 | 0.438 | 0.437 | **0.453** |
| PEMS-SF | 0.908 | 0.850 | 0.919 | 0.902 | 0.919 | **0.931** |
| *APDR* (%) | 1.785 | 4.599 | 6.288 | 7.183 | 5.116 | - |

on $K$ (*STCR w/o K*). We also compared the proposed method to two ablation models for loss function: STCR without instance-wise loss (*STCR w/o $\mathcal{L}^{IW}$*) and STCR without timestamp-wise loss (*STCR w/o $\mathcal{L}^{TW}$*). The accuracy scores of ablation models for STCR are listed in Table 3.

**Random cropping.** We compared the classification performance of STCR and *STCR w/o C*. Random cropping provides two subseries of an MTS instance with different lengths and positions. As presented in Table 3, *STCR w/o C* decreased approximately 1.785% compared to STCR on average for all datasets. Although the average performance drop rate was low compared to other ablation models, in some datasets, such as *JapaneseVowels* and *Heartbeat*, this model achieved a larger drop rate than others for context components; hence, it is one of the essential components of STCR.

**Random masking.** In general, most existing augmentation-based contrastive learning methods require a strong inductive bias, such as transformation invariance, that is not always suitable for handling MTS (Yue et al., 2022). Therefore, we only used random masking, a transformation that does not require strong assumptions. To verify the efficacy of random masking, we compared the classification performance of STCR and *STCR w/o M*. As presented in Table 3, *STCR w/o M* showed the classification performance of 4.599% decrease than STCR on average. Thus, we demonstrated that random masking can improve representation quality without unrealistic assumptions.

**Randomness on K.** We conducted three experiments to investigate the effects of *randomness on K* on classification performance, the ability to recognize long-term patterns, and the robustness to $\rho$.

First, we compared the classification performance of STCR and the ablation model trained by *STCR w/o K* to show the effect of the randomness on $K$. For the ablation model, we fixed $K$ to $0.5 \times V$ in the training phase. In Table 3, the classification performance of *STCR w/o K* was substantially decreased compared to that of STCR. Specifically, because random $K$ explicitly affects capturing the spatial structure of MTS, *STCR w/o K* showed the largest average drop rate of 6.288% compared to other ablation models in terms of context components.

Next, we verified that randomness on $K$ enhances the ability to recognize long-term patterns as well as the short-term patterns of MTS. As shown in Figure 4, STCR outperformed *STCR w/o K* on both short- and long-term forecasting tasks. Moreover, as the prediction length $H$ increased, the performance difference between STCR and *STCR w/o K* gradually increased in the ETTh2 dataset. Thus, we demonstrated that random $K$ can improve forecasting performance, even if the prediction length is long, by capturing inherent spatial information from the graphs with diverse $K$.

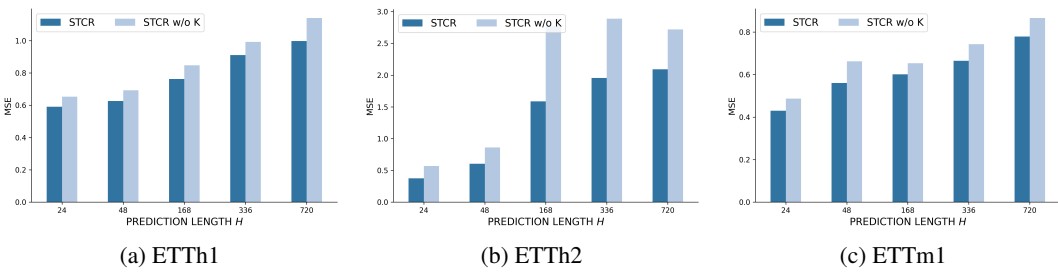

Figure 4: Difference of MSE between STCR and *STCR w/o K* on ETT datasets with various *H*s

We demonstrated the robustness against connection parameter $\rho$ used to determine the number of connections $K$ in the inference phase. Figure 5 shows the accuracy scores of STCR by varying $\rho \in [0.1, 0.9]$ (and the corresponding $K$). We observed that the classification performances for various $\rho$ values are similar for most datasets, which implies that STCR is not highly sensitive to $\rho$ owing to the randomness on $K$ in the training phase. Therefore, random $K$ enhances robustness to the number of connections, which may be challenging to set appropriately, by encouraging consistent representations for graphs consisting of various $K$s.

Thus, random $K$ improves classification and forecasting performances by reflecting inherent structural information for relationships between variables of MTS and enhancing robustness to $K$.

Figure 5: Accuracy of STCR across various connection parameters, $\rho$, (and corresponding $K$)

**Instance-wise contrastive loss.** As presented in Table 3, the classification performance of *STCR w/o $\mathcal{L}^{IW}$* was highly decreased for most datasets. Also, the ablation model showed a larger average performance drop rate (7.183%) than *STCR w/o $\mathcal{L}^{TW}$*. Thus, the instance-wise contrastive loss, $\mathcal{L}^{IW}$, fulfills a more important role for the classification task than $\mathcal{L}^{TW}$ by encouraging instances belonging to the same class to be close to each other with spatio-temporal consistency at the instance level.

**Timestamp-wise contrastive loss.** This loss function is useful for forecasting tasks by considering spatio-temporal consistency at the timestamp level. Table 4 provides the MSE and MAE results of two ablation models, *STCR w/o $\mathcal{L}^{IW}$* and *STCR w/o $\mathcal{L}^{TW}$*, and STCR on ETT datasets. On average, two ablation models

Table 4: MSE and MAE of ablation models. For each dataset, the lowest results are highlighted in boldface. (AEGR: Average error growth rate)

| Dataset | H | w/o $\mathcal{L}^{IW}$ | | w/o $\mathcal{L}^{TW}$ | | STCR | |
|---|---|---|---|---|---|---|---|
| | | MSE | MAE | MSE | MAE | MSE | MAE |
| ETTh1 | 24 | **0.588** | **0.542** | 0.688 | 0.584 | 0.591 | **0.542** |
| | 48 | 0.628 | 0.569 | 0.730 | 0.611 | **0.626** | **0.565** |
| | 168 | 0.766 | 0.650 | 0.877 | 0.692 | **0.762** | **0.644** |
| | 336 | 0.955 | 0.745 | 1.024 | 0.766 | **0.910** | **0.721** |
| | 720 | 1.034 | 0.792 | 1.157 | 0.841 | **0.997** | **0.770** |
| ETTh2 | 24 | **0.368** | **0.462** | 0.434 | 0.481 | 0.375 | 0.467 |
| | 48 | 0.616 | 0.595 | 0.657 | 0.610 | **0.605** | **0.597** |
| | 168 | 1.693 | 1.016 | 1.726 | 1.017 | **1.587** | **1.002** |
| | 336 | 2.614 | 1.264 | 2.652 | 1.288 | **1.956** | **1.130** |
| | 720 | 2.139 | 1.177 | 2.330 | 1.189 | **2.092** | **1.159** |
| ETTm1 | 24 | 0.463 | 0.465 | 0.484 | 0.460 | **0.430** | **0.432** |
| | 48 | 0.612 | 0.552 | 0.638 | 0.547 | **0.560** | **0.514** |
| | 96 | 0.642 | 0.573 | 0.652 | 0.572 | **0.601** | **0.545** |
| | 288 | 0.720 | 0.619 | 0.745 | 0.630 | **0.665** | **0.589** |
| | 672 | 0.818 | 0.676 | 0.888 | 0.699 | **0.779** | **0.654** |
| AEGR (%) | | 8.274 | 3.543 | 15.854 | 6.350 | - | - |

showed worse forecasting performance than STCR. Specifically, although STCR showed slightly superior performance than ablation models on short-term forecasting, STCR achieved overwhelming performance for long-term forecasting. Moreover, *STCR w/o $\mathcal{L}^{TW}$* exhibited a higher average error growth rate than *STCR w/o $\mathcal{L}^{IW}$* did. This result implies that the instance-wise and timestamp-wise contrastive loss functions can improve forecasting performance as well as classification performance; in addition, the timestamp-wise contrastive loss enables the model to effectively generate representations more suitable for forecasting tasks than the instance-level contrastive loss.

Table 5: Accuracy scores of ablation models against consistency along with STCR. For each dataset, the best score is highlighted in boldface.

| Dataset | STCR | → Subseries | → Local | → Jittering | → Fliping | → Permutation |
|---|---|---|---|---|---|---|
| HandMovementDirection | 0.351 | **0.405** | 0.257 | 0.257 | 0.230 | 0.311 |
| PhonemeSpectra | 0.186 | 0.129 | 0.109 | 0.180 | 0.186 | **0.199** |
| JapaneseVowels | **0.989** | 0.968 | 0.981 | 0.968 | 0.981 | 0.976 |
| SpokenArabicDigits | **0.955** | 0.940 | 0.910 | 0.953 | 0.921 | 0.937 |
| NATOPS | **0.944** | 0.933 | 0.828 | 0.894 | 0.928 | 0.894 |
| FingerMovements | **0.590** | 0.560 | 0.530 | 0.450 | 0.570 | **0.590** |
| Heartbeat | **0.746** | **0.746** | 0.732 | 0.741 | 0.722 | 0.722 |
| MotorImagery | **0.550** | 0.490 | 0.500 | 0.540 | 0.500 | 0.500 |
| FaceDetection | 0.557 | **0.558** | 0.547 | 0.548 | 0.537 | 0.543 |
| InsectWingbeat | **0.453** | 0.417 | 0.365 | 0.445 | 0.446 | 0.434 |
| PEMS-SF | **0.931** | 0.908 | 0.896 | 0.861 | 0.902 | 0.844 |
| *Average* | **0.659** | 0.641 | 0.605 | 0.622 | 0.629 | 0.632 |

**Spatio-temporal consistency.** Furthermore, to demonstrate the effectiveness of spatio-temporal consistency, we performed additional ablation studies against the proposed spatio-temporal consistency. We compared the classification performance of our method to those of five ablation models: two STCRs replacing the proposed spatio-temporal consistency with subseries and local consistencies, respectively, and three STCRs replacing random masking with jittering, flipping, and permutation, respectively. Consequently, as shown in Table 5, the proposed spatio-temporal consistency outperformed the ablation models in most datasets and achieved the best average accuracy.

## 5 CONCLUSION

We proposed a novel representation learning method, STCR, to learn universal representations for various tasks of MTS by encouraging spatio-temporal consistency to capture inherent spatial information and temporal dependency while mitigating the drawbacks of the typical selection strategies. We obtained two context views using random cropping, temporal embedding, spatial embedding, and projection modules; a spatio-temporal representation is learned by instance-wise and timestamp-wise contrastive loss functions encouraging spatio-temporal consistency between two context views. Through extensive experiments on classification and forecasting tasks, we demonstrated that STCR is useful for generating universal representations while performing better than SOTAs.

**Limitations.** The proposed method has two limitations. First, STCR shows relatively low performance in some datasets, especially for those with a small number of variables. Because STCR reflects structural relations among the variables in MTS by a $K$-NN graph, the performance for the datasets with few variables or no meaningful spatial information can be suboptimal. Second, although STCR shows comparable processing times with TS2Vec, the SOTA with high efficiency in MTS representation learning, in a reasonable number of variables, the processing time rapidly increases with the number of variables (see Table 6). For example, in *PEMS-SF* dataset, which has 963 variables, STCR remarkably improved classification performance by about 36% compared to TS2Vec by reflecting spatial information (see Table 1). However, in terms of computational time, since the $K$-NN graph constructed by these variables requires a large amount of additional computation in learning the graph neural network and processing with the learned network in the inference phase ($O(V^2)$ where $V$ is the number of variables or nodes), STCR performed slower than TS2Vec.

**Future research directions.** One possible solution to handle MTS datasets with a small number of variables using our approach is to construct $K$-NN graphs in the latent space formed by simple embedding. If the latent representations contain spatial information on structural relations of the input MTS data, we expect to gain an effect similar to our approach. Beside, by constructing graphs with latent representations, STCR can also be adapted to UTS. Meanwhile, to enhance the superiority of our method by reducing the complexity, we can devise an efficient method that simplifies a large graph to a small graph without losing spatial information.

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

# A  IMPLEMENTATION DETAILS

## A.1  PREPROCESSING

Following previous works (Franceschi et al., 2019; Zhou et al., 2021; Yue et al., 2022), we used three preprocessing techniques to handle MTS:

**Normalization of variables with different scales.**    Because each variable in MTS has a different scale, we normalized each variable independently using a z-score. In forecasting tasks, we measured all evaluation metrics with normalized values.

**Handling variable-length and missing values.**    For a variable-length dataset, we padded all instances to have the same length by *NaNs*, meaning the missing values. When missing values occurred, we masked the corresponding positions as zero. Moreover, when we handled the graph of subseries in the spatial embedding module, we padded the node features to the same dimension as the sequence length of the input instance by zero.

**Use of timestamp information.**    We used additional timestamp features, including minute, hour, day-of-week, day-of-month, day-of-year, month-of-year, and week-of-year, when available.

## A.2  HYPERPARAMETERS

Because most previous studies for representation learning assume that label information and downstream tasks are unknown, selecting appropriate hyperparameters based on the model performance is difficult. Hence, following Yue et al. (2022), we used fixed hyperparameters regardless of the downstream tasks and did not perform additional hyperparameter optimization. Referring to (Yue et al., 2022), the batch size $B$ was set to 8, and the learning rate was $10^{-3}$. The number of optimization iterations was set to 200 for datasets when the number of instances was less than 100,000; otherwise, it was set to 600. In the training phase, when the instance has a sequence length larger than 3,000, we clipped the sequence into segments with 3,000 timestamps. Encoder $f$ for the temporal embedding module contained ten hidden temporal blocks consisting of two *DilatedConv* with an activation function, *GeLU* (Hendrycks & Gimpel, 2016), and skip connections existed between neighboring blocks. For the $\ell$-th block, the dilation parameter was set to $2^{\ell}$. The kernel size was set to 3, each *DilatedConv* had a dimension of 64, and a residual block mapped the hidden features to $d_{z^{\mathcal{T}}}$-dimensional temporal features. We set $p$ in the Bernoulli distribution for random masking to 0.5. Subsequently, the encoder $g$ used in the spatial embedding module was configured with three hidden spatial blocks consisting of *GraphConv* and *GeLU*. We set the output dimensions of *Graph-Conv*s corresponding to each spatial block to 128, 64, and $d_{z^s}$, respectively. In addition, for the spatial embedding module, we set the connection parameter $\rho$ to 0.5 in the training phase because low-quality graphs with insufficient information for relationships between variables of MTS hinder obtaining effective spatial features; thereby, the range of $K$ was $[\lceil 0.5 \times V \rceil, V]$. In the inference phase, we fixed $K$ to the median value of the range of $K$ used in the training phase. In the projection module, we configured $h$ with two fully connected layers with 64 and $d_z$ dimensions, respectively. Here, $d_{z^{\mathcal{T}}}$, $d_{z^s}$, and $d_z$ were equally set to 320.

All experiments were executed on the Pytorch platform using an Intel Core i9-10900X at 3.70 GHz CPU, 256 GB RAM, and GeForce RTX 3090 24GB GPU. The code for STCR is implemented based on the official code of TS2Vec [1]. Our code is attached as a zip file in submission.

# B  EXPERIMENTAL SETTINGS

## B.1  MULTIVARIATE TIME-SERIES CLASSIFICATION

For classification tasks, a class should be assigned to each MTS instance; hence, instance-level representations are required. For a fair comparison, we used max pooling over all timestamps to obtain the instance-level representations following Yue et al. (2022). Then, following the same

---

[1]https://github.com/yuezhihan/ts2vec

protocol with Franceschi et al. (2019) and Yue et al. (2022), we trained a support vector machine (SVM) classifier with the RBF kernel using the instance-level representations to predict the class of each instance. We set the penalty $C$ with a grid search ranging in $[10^{-4}, 10^4]$ by cross-validation for the training dataset.

**Datasets.** We used MTS classification datasets from the University of East Anglia and the University of California Riverside (UEA & UCR) time-series classification repository for evaluation. Among 30 MTS datasets in the repository, we selected 11 datasets that have at least ten variables and 100 training instances. The repository separately provides training and test datasets, so we used the training dataset to train the model and the test dataset to evaluate the trained model.

**Baselines.** To demonstrate that the representations learned by STCR are suitable for the classification task, we compared the proposed method to SOTAs in unsupervised time-series representation learning, including T-Loss (Franceschi et al., 2019), TS-TCC (Eldele et al., 2021), TST (Zerveas et al., 2021), TNC (Tonekaboni et al., 2021), and TS2Vec (Yue et al., 2022) in addition to DTW (Chen et al., 2013). Each method is summarized in the main manuscript. We used the results reported in Yue et al. (2022) for all baseline methods.

**Evaluation metric.** We evaluated the classification performance by measuring the accuracy score.

## B.2 MULTIVARIATE TIME-SERIES FORECASTING

Given the last $T$ observations $x_{t-T+1}, \cdots, x_t$, we predict the $H$ upcoming observations $x_{t+1}, \cdots, x_{t+H}$ by using $z_t$, the spatio-temporal representation of the last timestamp $t$. Specifically, we adopted a protocol of Yue et al. (2022), which used a linear regression model with $L_2$ regularization trained by using $z_t$ as the input to directly predict future observations $\hat{x}_{t+1}, \cdots, \hat{x}_{t+H}$. We set the regularization coefficient $\alpha$ by a grid search on the validation dataset from search space $\{0.1, 0.2, 0.5, 1, 2, 5, 10, 20, 50, 100, 200, 500, 1000\}$.

**Datasets.** To evaluate forecasting performance, we used four public datasets, including three ETT datasets (Zhou et al., 2021) and Electricity dataset (Dua et al., 2017). ETT datasets, including ETTh1, ETTh2, and ETTm1, collect two years of power transformer data, containing long-term trends, periodicity, and irregular patterns from two stations. ETTh1 and ETTh2 were collected every hour, and ETTm1 was collected in a 15-min unit. The Electricity dataset contained the electricity consumption data for 321 clients over three years. Following Zhou et al. (2021) and Yue et al. (2022), we resampled the Electricity data into hourly data. In addition, we split ETT datasets into training, validation, and test datasets with 12, 4, and 4 months, respectively (Zhou et al., 2021). The Electricity was split into 60%, 20%, and 20% (Yue et al., 2022). To demonstrate the performance of both short- and long-term forecasting, we lengthened the prediction length $H$ progressively, from 1 day to 30 days for hourly data and from 6 hours to 7 days for minute data.

**Baselines.** We compared the forecasting performance of STCR with TS2Vec, which is the SOTA of MTS representation learning. In addition, we also compared our method with the SOTAs of MTS forecasting tasks, including Informer (Zhou et al., 2021), LogTrans (Li et al., 2019), LSTnet (Lai et al., 2018), TCN (Bai et al., 2018), and StemGNN (Cao et al., 2020).

- **Informer** is a transformer-based method for efficiently forecasting MTS.
- **LogTrans** improves forecasting performance by mitigating the memory bottleneck of the transformer.
- **LSTnet** uses both convolutional and recurrent neural networks to recognize both short-term and long-term trends of MTS.
- **TCN** is an MTS forecasting method that introduces *DilatedConv* for the first time.
- **StemGNN** is an MTS forecasting method that uses spectral information with Fourier transform to improve forecasting performance, whereas most other spatio-temporal graph neural networks focusing on the graphs may differ in performance by how to construct graphs from time series.

We used the results reported in Yue et al. (2022) for all baseline methods.

**Evaluation metrics.** We used two evaluation metrics for forecasting tasks: mean squared error (MSE) and mean absolute error (MAE). MSE is measured as follows:

$$MSE = \frac{1}{HV} \sum_{h=1}^{H} \sum_{v=1}^{V} (x_{t+h}^v - \hat{x}_{t+h}^v)^2, \tag{10}$$

where $x_{t+h}^v$ and $\hat{x}_{t+h}^v$ are the observed and predicted values on variable $v$ at timestamp $t + h$, respectively. Another metric, MAE, is measured as follows:

$$MAE = \frac{1}{HV} \sum_{h=1}^{H} \sum_{v=1}^{V} |x_{t+h}^v - \hat{x}_{t+h}^v|. \tag{11}$$

## C ADDITIONAL EXPERIMENTS

### C.1 COMPLEXITY ANALYSIS

Since our encoder architecture has an additional embedding module to handle the $K$-NN graph compared to TS2Vec (Yue et al., 2022), which is the SOTA with high efficiency in MTS representation learning, it requires more computation. In this regard, here, we discuss the efficiency of the proposed method from two perspectives: the number of parameters and computational time.

To examine the number of learnable parameters of our method, we denoted the number of variables passing through $l$-th layer of the temporal encoder as $V^{(l)}$, and the sequence length after $l'$-th layer of the spatial encoder as $L_x^{(l')}$. $L$ and $L'$ are the number of layers in the temporal and spatial encoders, respectively. In addition, the size of the one-dimensional convolution filter in the temporal encoder is denoted as $d_k$. Thus, the STCR has the following number of learnable parameters:

$$\sum_{l=1}^{L} V^{(l-1)}V^{(l)}d_k + \sum_{l'=1}^{L'} L_x^{(l'-1)}L_x^{(l')} + (V^{(L)} + L_x^{(L')})d_h + d_h d_z, \tag{12}$$

where $V^{(0)}$ and $L_x^{(0)}$ are the number of variables and sequence lengths of the original time series, respectively. In addition, $d_h$ is the number of hidden features in the projection head. The first and second terms are the number of learnable parameters of the temporal and spatial encoders, respectively, and the others refer to the number of parameters in the projection head consisting of two fully connected layers.

By contrast, TS2Vec does not consider spatial information of multivariate time series, so it has the following number of learnable parameters:

$$\tilde{V}V^{(0)} + \sum_{l=1}^{L} V^{(l-1)}V^{(l)}d_k \tag{13}$$

where $\tilde{V}$ is the number of variables in the time series, and $V^{(0)}$ is the number after the input projection layer. The first term indicates the number of parameters in the input projection layer, and the second term regards the rest parts of the encoder.

Thus, the total number of learnable parameters for TS2Vec and STCR differs approximately as much as the number of learnable parameters required by the spatial encoder of STCR.

Then, we also compared the processing time of our method to that of TS2Vec. For two methods, we reported the training time per iteration and the inference time per instance in Table 6. The proposed method shows comparable processing times with TS2Vec with a reasonable

Table 6: Training time per iteration and inference time per instance for STCR and TS2Vec

| Dataset | TS2Vec | | STCR | |
|---|---|---|---|---|
| | Training | Inference | Training | Inference |
| HandMovementDirection | 0.223 | 6.89e-4 | 0.209 | 2.86e-3 |
| PhonemeSpectra | 0.193 | 2.02e-4 | 0.184 | 2.45e-3 |
| JapaneseVowels | 0.059 | 2.46e-4 | 0.062 | 2.53e-3 |
| SpokenArabicDigits | 0.158 | 1.73e-4 | 0.167 | 2.48e-3 |
| NATOPS | 0.148 | 1.94e-4 | 0.174 | 2.78e-3 |
| FingerMovements | 0.148 | 2.00e-4 | 0.180 | 2.91e-3 |
| Heartbeat | 0.227 | 3.85e-4 | 0.349 | 5.65e-3 |
| MotorImagery | 0.508 | 1.13e-3 | 0.562 | 8.36e-3 |
| FaceDetection | 0.151 | 2.01e-4 | 1.167 | 2.65e-2 |
| InsectWingbeat | 0.138 | 2.07e-4 | 2.133 | 6.33e-2 |
| PEMS-SF | 0.178 | 5.61e-4 | 45.91 | 1.09e-0 |

number of variables, but the processing time rapidly increases with the number of variables. For example, in the PEMS-SF dataset, which has 963 variables, STCR remarkably improved classification performance by about 25 percentage points (or 36.5%) compared to TS2Vec by reflecting spatial information (see Table 1 in the manuscript). However, in terms of computational time, since the graph constructed by these variables requires a large amount of additional computation in learning the graph neural network and processing with the learned network in the inference phase ($O(V^2)$) where $V$ is the number of variables or nodes), STCR can be slower than TS2Vec.

## C.2 GRAPHICAL ANALYSIS

We performed a graphical analysis of the representations learned by the proposed method using UMAP (McInnes et al., 2018), a recently proposed technique for visualization. We selected two datasets, PEMS-SF and JapaneseVowels, which exhibit the largest and smallest performance gaps, respectively, between STCR and TS2Vec, which is the SOTA in MTS representation learning.

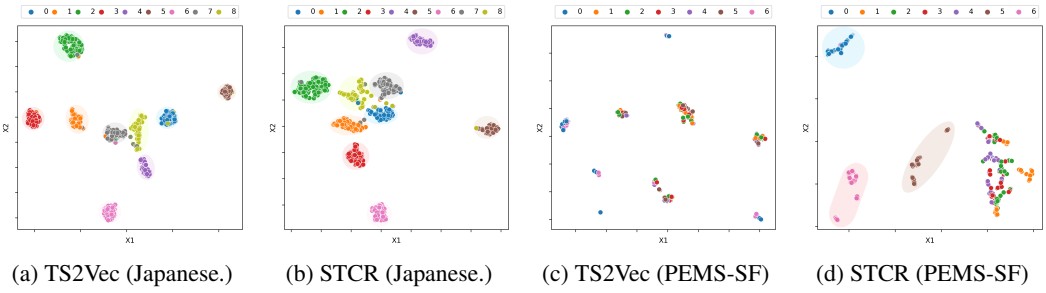

| (a) TS2Vec (Japanese.) | (b) STCR (Japanese.) | (c) TS2Vec (PEMS-SF) | (d) STCR (PEMS-SF) |

Figure 6: Visualization on JapaneseVowels ((a) and (b)) and PEMS-SF ((c) and (d))

As shown in Figure 6, for the JapaneseVowels dataset, we observed that both methods form well-distinguished clusters for all classes. However, for the PEMS-SF dataset, the representations learned by STCR form significantly better distinct groups for each class than TS2Vec, especially the classes 0, 5, and 6. Through this analysis, we can reaffirm the effectiveness of the proposed method.

## C.3 SENSITIVITY ANALYSIS

To investigate the impact of hyperparameters used in our method, we performed sensitivity analyses for the number of optimization iterations $Q$ and the assigned weights for two loss functions, $\mathcal{L}^{IW}$ and $\mathcal{L}^{TW}$. Here, we followed the experimental settings described in Section B.1.

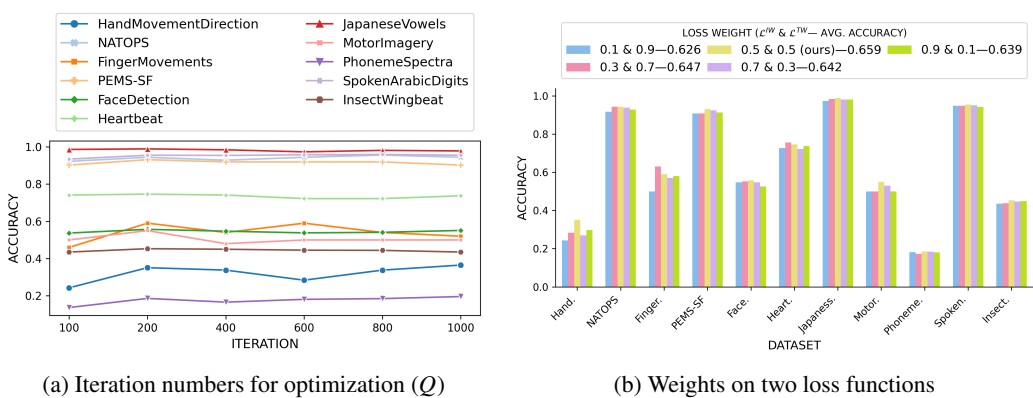

| (a) Iteration numbers for optimization ($Q$) | (b) Weights on two loss functions |

Figure 7: Accuracy scores of STCR (a) across various iteration numbers for optimization and (b) when assigning different weights for two loss functions

**Iteration numbers for optimization.** We observed the classification performance of the proposed method on 11 MTS datasets by varying optimization iterations, $Q$. As shown in Figure 7(a), in most datasets, the classification performance gradually increased until about 200 iterations, whereas slightly decreasing or being stable after that.

**Weights on two loss functions.** We compared the classification performances of STCR on 11 MTS datasets when assigning different weights for instance-wise and timestamp-wise contrastive losses, $\mathcal{L}^{IW}$ and $\mathcal{L}^{TW}$. As shown in Figure 7(b), although assigning the same weights achieved the best average accuracy, we show that our approach is not sensitive to the weights in most datasets.

## D    NOTATIONS

For a comprehensive understanding of our work, we provided a notation table, including the dimensions of the variables, in Table 7.

Table 7: Notations

| Notation | Description | Dimension |
|---|---|---|
| $\mathcal{X}$ | Set of multivariate time-series instances | - |
| $N$ | Number of instances | - |
| $x_i$ | Multivariate time-series instance | $L_x \times V$ |
| $L_x$ | Sequence length of $x_i$ | - |
| $V$ | Number of variables | - |
| $\tilde{x}_i$ | Randomly extracted subseries | $L_{\tilde{x}_i} \times V$ |
| $L_{\tilde{x}_i}$ | Sequence length of $\tilde{x}_i$ | - |
| $\tilde{x}'_i$ | Another randomly extracted subseries | $L_{\tilde{x}'_i} \times V$ |
| $L_{\tilde{x}'_i}$ | Sequence length of $\tilde{x}'_i$ | - |
| $f$ | Temporal encoder | - |
| $z_i^{\mathcal{T}}$ | Temporal features of $x_i$ | $L_x \times d_{z^{\mathcal{T}}}$ |
| $\tilde{z}_i^{\mathcal{T}}$ | Temporal features of $\tilde{x}_i$ | $L_{\tilde{x}_i} \times d_{z^{\mathcal{T}}}$ |
| $m_i$ | Binary mask $\in \{0, 1\}$ of $\tilde{x}_i$ | $L_{\tilde{x}_i}$ |
| $g$ | Spatial encoder | - |
| $z^{\mathcal{S}}$ | Spatial features of $x_i$ | $L_x \times d_{z^{\mathcal{S}}}$ |
| $\tilde{z}_i^{\mathcal{S}}$ | Spatial features of $\tilde{x}_i$ | $L_{\tilde{x}_i} \times d_{z^{\mathcal{S}}}$ |
| $\zeta_i^{(v,u)}$ | Similarity between variables $v$ and $u$ of $\tilde{x}_i$ | - |
| $\rho$ | Connection parameter | - |
| $K$ | Number of connections | - |
| $\mathcal{G}_i^K$ | Graph with $K$ connections of $\tilde{x}_i$ | - |
| $A_i^K$ | Adjacency matrix of $\mathcal{G}_i^K$ | $V \times V$ |
| $\bar{A}_i^K$ | Adjacency matrix with self-loop ($\bar{A}_i^K = A_i^K + I$) | $V \times V$ |
| $\bar{D}$ | Degree matrix of $\bar{A}_i^K$ | $V \times V$ |
| $I$ | Identity diagonal matrix of $A_i^K$ | $V \times V$ |
| $\mathcal{S}_i$ | Spatial information of $\tilde{x}_i$ | $V \times d_{z^{\mathcal{S}}}$ |
| $h$ | Projection head | - |
| $z_i$ | Spatio-temporal representation of $x_i$ | $L_x \times d_z$ |
| $\tilde{z}_i$ | Spatio-temporal representation of $\tilde{x}_i$ | $L_{\tilde{x}_i} \times d_z$ |
| $\tilde{z}'_i$ | Spatio-temporal representation of $\tilde{x}'_i$ | $L_{\tilde{x}'_i} \times d_z$ |
| $\mathcal{L}^{IW}$ | Instance-wise contrastive loss | - |
| $\mathcal{L}^{TW}$ | Timestamp-wise contrastive loss | - |
| $\mathcal{L}$ | Overall loss | - |
| $B$ | Batch size | - |
| $\Phi$ | Set of the overlapping timestamps of two subseries | - |

