# OpenReview forum: "Learning Spatio-Temporal Representation for Multivariate Time Series"
_ICLR.cc/2024/Conference — ICLR 2024 Conference Withdrawn Submission_

### Official Review · Reviewer_Egy1 · 2023-10-15

**Soundness:** 1 poor
**Presentation:** 3 good
**Contribution:** 1 poor
**Rating:** 3
**Confidence:** 5

**Summary:**

This paper proposes a spatio-temporal time-series representation learning framework that incorporates two encoders for temporal aspect and cross-variable aspect, respectively.

**Strengths:**

1. Very clear writing and illustrations by simple figures.
2. Extensive experiments using more than 10 datasets in each task.

**Weaknesses:**

1. STCR is not novel and seems to be a simple modification of TS2Vec by adding k-NN graph construction and GCN architecture.

2. No consideration about selecting positive pairs, repeating what have done in TS2Vec. The positive pairs are still sampled from the overlapped region between two sampled window. The fundamental variation in positive pair sampling is not accomplished.

3. No enough connection between coping with distribution changes and the modules of STCR. There is no explanation about how STCR can deal with distribution changes.

**Questions:**

1. Why does the paper title contain the term "spatial"? There is no geographical dataset and the method does not exploit geographical aspect, just modeling cross-variable correlations.

2. The term "structure" in Introduction is ambiguous.

---

### Official Review · Reviewer_rtxb · 2023-10-26

**Soundness:** 2 fair
**Presentation:** 3 good
**Contribution:** 2 fair
**Rating:** 3
**Confidence:** 4

**Summary:**

The paper introduces a multi-view contrastive learning framework for acquiring effective multivariate time series representations. While the proposed methodology appears technically sound, it fails to provide groundbreaking insights to benefit broader research. Some assertions and technical facets within this work are not thoroughly substantiated. The experiments, though fairly exhaustive, could benefit from further refinement.

**Strengths:**

1. The paper is well-organized and generally well-written.
2. The primary motivation for self-supervised time series representation is clearly articulated.
3. The introduced spatio-temporal consistency optimization appears logical, and the experimental results effectively showcase its efficacy.

**Weaknesses:**

1. The technical novelty of the paper is somewhat constrained. The proposed approach bears notable resemblances to TS2Vec in aspects such as random cropping, temporal encoding, and two contrastive losses. The main distinction is in the introduced spatial consistency.
2. Several claims and model design choices lack thorough discussion or theoretical backing. A case in point is the spatial consistency proposal. Firstly, this design is tightly coupled with the underlying graph structure learning choice, making it less general and flexible. Secondly, its necessity is debatable, especially given recent research that indicates channel independence is often effective in modeling MTS data [1,2]. Additionally, the paper doesn't show significant performance improvements over TS2Vec in the majority of the forecasting endeavors. Further elaborations on these points can be found in the subsequent comments and questions I've listed.
3. The paper's technical writing could be more polished, and it omits comparisons with some crucial and relevant baselines [1,2,3]. Based on the data provided, the assertion of STCR generating top-tier representations and leading performance across tasks isn't convincingly supported.

[1] Nie, Y., Nguyen, N. H., Sinthong, P., & Kalagnanam, J. (2022). A time series is worth 64 words: Long-term forecasting with transformers. arXiv preprint arXiv:2211.14730.

[2] Dong, J., Wu, H., Zhang, H., Zhang, L., Wang, J., & Long, M. (2023). SimMTM: A Simple Pre-Training Framework for Masked Time-Series Modeling. arXiv preprint arXiv:2302.00861.

[3] Wu, H., Hu, T., Liu, Y., Zhou, H., Wang, J., & Long, M. (2022). Timesnet: Temporal 2d-variation modeling for general time series analysis. arXiv preprint arXiv:2210.02186.

**Questions:**

**Questions & Detailed comments**

1. The emphasis on labeling information in time series (e.g., "... using MTS label information in practice is challenging because MTS is sometimes sparsely labeled" at the start of the introduction) seems misplaced, as such labeling is often irrelevant in primary forecasting tasks.
2.  Regarding the assertion in the third paragraph of the introduction, I question the necessity to model along the variable/spatial dimension as previously noted. Also, when claiming, "Therefore, the existing methods that do not consider spatial information cannot achieve high performance in tasks for MTS compared with those for univariate time series (UTS)", I don't observe a marked enhancement of this work over TS2Vec.
3. The statement at the start of Sec 3.2, namely "... while alleviating the drawbacks of three typical selection strategies for constructing positive pairs", lacks a supporting discussion.
4. Fig 1 lacks clarity. Specifically, what is meant by the output of "random cropping"? Is it \tilde{x}_i, \tilde{x}'_i, or both?
5. Concerning random cropping, how is the overlapping length determined? Also, the assertion, "this approach enables us to learn position-agnostic representations while avoiding dimension collapse", does not seem to be backed by any evidence.
6. Why we have to use this KNN-based method to shape the graph topology in the context of spatial consistency? It seems that the proposed spatial consistency heavily relies on this assumption, particularly when attempting "to recognize the underlying spatial structure of MTS by addressing this problem, we encouraged the representations of K-NN graphs to be consistent across various K". Note that there are multiple ways in building the graph structures for time series.
7. In reference to the overall framework and Eq.6, why opt to concatenate two embeddings instead of integrating the temporal and spatial modules to represent a time series?
8. Algorithm 1, as presented, doesn't appear to add value and might be omitted without affecting comprehension.
9. In the experiments, what are the classification and forecasting heads? Also, why were results for ETTm2 not presented?

---

### Official Review · Reviewer_Rr7Y · 2023-10-29

**Soundness:** 3 good
**Presentation:** 2 fair
**Contribution:** 3 good
**Rating:** 6
**Confidence:** 3

**Summary:**

This paper presents a method called Spatio-Temporal Contrastive Representation Learning (STCR) for generating effective representations of Multivariate Time Series (MTS). The paper alleviates the limitations of existing contrastive learning methods and proposes a solution that considers spatio-temporal information and temporal dependency. The experimental results demonstrate the effectiveness of STCR in achieving high-quality representations and outperforming state-of-the-art methods in MTS classification and forecasting tasks.

**Strengths:**

1. The proposed method addresses the limitations of existing contrastive learning methods for sequential data and provides a comprehensive solution for MTS representation learning.
2. The experimental results show that STCR outperforms state-of-the-art methods in MTS classification and forecasting tasks, indicating the effectiveness of the proposed method.

**Weaknesses:**

1. The paper lacks a detailed analysis of the computational complexity and scalability of the proposed method. It would be beneficial to discuss the efficiency of STCR compared to other methods.
2. The paper could provide more insights into the interpretability of the learned representations and how they capture the spatio-temporal information in MTS.
3. The paper lacks an analysis of why STCR is not as effective as other models in some datasets such as Heartbeat.

**Questions:**

From the subsection 'Random cropping', the two context views are of different lengths and are not aligned in time. But from the loss defined in Equ. 7 and 8, it seems that the two views are of the same length and aligned.

---

### Official Review · Reviewer_eaUW · 2023-10-30

**Soundness:** 3 good
**Presentation:** 3 good
**Contribution:** 2 fair
**Rating:** 5
**Confidence:** 4

**Summary:**

This paper proposes a contrastive learning framework to generate robust representations of multivariate time series, named STCR. STCR applied four modules, i.e., random cropping, temporal embedding, spatial embedding and projection, to generate representation of MTS instances. The four modules are commonly used in existing works and the contribution of this work is insignificant.

**Strengths:**

1. The writing of this paper is good and the whole framework is easy to follow.
2. Comprehensive experiments are conducted on both MTS classification and forecasting tasks.
3. Implementation details are given in appendix, which is appreciated and makes the result more persuasive.

**Weaknesses:**

1. The proposed framework lacks novelty. The motivation of this paper lies in 1) Spatial information is not considered. However, many works considering spatial information, e.g., [1]. 2) Existing strategies of constructing positive pairs are vulnerable to distribution shifts. But the authors fail to clarify how the proposed framework can be stable or robust to TDS.
2. Lacking recent baselines in experiments, e.g., [2]. And the performance of the proposed framework is not that good.
3. The four modules in the frameworks are just combination of existing technics and the contribution of this work is unclear.
4. The motivation of the design of the four modules should be further clarified.


[1] Wu Z, Pan S, Long G, et al. Connecting the dots: Multivariate time series forecasting with graph neural networks[C]//Proceedings of the 26th ACM SIGKDD international conference on knowledge discovery & data mining. 2020: 753-763.
[2] Luo D, Cheng W, Wang Y, et al. Time series contrastive learning with information-aware augmentations[C]//Proceedings of the AAAI Conference on Artificial Intelligence. 2023, 37(4): 4534-4542.

**Questions:**

See weaknesses.

---

### Official Review · Reviewer_3yDx · 2023-11-01

**Soundness:** 2 fair
**Presentation:** 3 good
**Contribution:** 2 fair
**Rating:** 3
**Confidence:** 4

**Summary:**

The authors aimed to address two problems when applying contrastive learning to time-series data. First, time-series data inherently exhibits spatial-temporal information while existing methods ignored that. Second, existing augmentation methods for time-series ignored the property of distribution changes. To address these problems, the authors proposed a method called spatial-temporal contrastive representation learning (STCR). Specifically, STCR achieved spatial-temporal consistency for time-series data and addressed the drawbacks of conventional augmentation methods.

**Strengths:**

Experiments are extensive and the motivations are interesting.

**Weaknesses:**

1. This work lacks novelty. The overall structure is similar to TS-TCC [1] and TS2Vec [2]. For example, compared to TS2Vec, this manuscript is just adding GCN to consider spatial dependencies within MTS data.

[1] Time-Series Representation Learning via Temporal and Contextual Contrasting

[2] TS2Vec: Towards Universal Representation of Time Series

2. References are too old, and the authors have done limited literature review. There are works highly related to this work. For example, GCC [3] also conducted graph neural network and explore the effects of spatial consistency. I think the authors should discuss their differences.

[3] Graph Contextual Contrasting for Multivariate Time Series Classification

3. Contributions have not addressed the problems that the authors pointed out. The authors pointed out the limitation in the existing augmentation methods, i.e., existing methods cannot consider the distribution changes in time-series data. To address the limitation, the authors proposed a method namely random cropping. I cannot see the difference between the random cropping and existing method subseries. Meanwhile, I don't think that the proposed method can address the mentioned problem. In the example of Figure 2, the signals within [s1, s2] are still having different distribution with [e1, e2]. Following the point of the authors (changing distribution), the two subseries cannot be treated as positive pairs.

4. The authors mentioned the importance of achieving spatial-temporal consistency. However, how to define the spatial consistency? Why can spatial consistency be achieved after you construct and process a K-NN graph with GCN?

**Questions:**

1. What is the differences between this work and existing studies, such as GCC [3].

[3] Graph Contextual Contrasting for Multivariate Time Series Classification

2. Please clarify the reason why random cropping can address the problem of distribution changes.

3. How to define spatial consistency, and why can spatial consistency be achieved after you construct and process a K-NN graph with GCN?